# Application of Solvent Evaporation to Generate Supersaturated Lipid-Based Formulations: Investigation of Drug Load and Formulation Quality

**DOI:** 10.3390/pharmaceutics17060702

**Published:** 2025-05-27

**Authors:** Felix Paulus, Jef Stappaerts, Annette Bauer-Brandl, Dirk Lauwers, Liesbet Smet, Eline Hermans, René Holm

**Affiliations:** 1Janssen Pharmaceutica NV, Turnhoutseweg 30, 2340 Beerse, Belgiumjstappa1@its.jnj.com (J.S.); dlauwers@its.jnj.com (D.L.); lsmet@its.jnj.com (L.S.); eherman7@its.jnj.com (E.H.); 2Department of Physics, Chemistry and Pharmacy, University of Southern Denmark, Campusvej 55, 5230 Odense, Denmark; annette.bauer@sdu.dk

**Keywords:** supersaturated lipid-based formulations, lipid-based drug delivery systems, solubility, supersaturation, residual solvents

## Abstract

**Background/Objectives:** Lipid-based formulations (LBFs) are enabling formulations for poorly water-soluble, mostly lipophilic drugs. In LBFs, the drug is pre-dissolved in the formulation which can consist of lipids, surfactants, and/or cosolvents. In cases where the administration of high amounts of a drug is required, exceeding the drug solubility in the lipidic vehicle at the administration temperature, supersaturated LBFs are an option. The standard method described in the literature for inducing supersaturation in LBFs is to dissolve the drug substance in the lipidic vehicle at an elevated temperature, e.g., at 60 °C, and then subsequently let the formulation cool to ambient temperature before administration (heat-based approach). In this work, an alternative approach to induce supersaturation in LBFs was investigated in order to evaluate if higher drug loads, i.e., the concentration of drug dissolved in the vehicle, could be reached compared to the loading obtainable via heating. **Methods:** A volatile solvent that is miscible with the lipid matrix and in which the compound has a high solubility is added to the lipid matrix, after which the solvent is evaporated. Both approaches were compared in this work investigating two different LBFs loaded with the BCS-class II drugs celecoxib and fenofibrate. **Results:** When inducing supersaturation by heat, drug loads of 238% for celecoxib and 278% for fenofibrate could be achieved relative to the solubility at ambient temperature. Using the solvent-based approach, drug loads of up to 475% for celecoxib and 557% for fenofibrate could be prepared in the LBFs using dichloromethane (DCM) as the volatile solvent. However, those highly supersaturated preparations showed suboptimal physical stability and quickly led to precipitation when the LBFs were stored at ambient temperature. In addition, selected formulations were analyzed with GC-headspace to determine the residual DCM after solvent evaporation using a vacuum evaporator. This analysis revealed that the DCM content exceeded regulatory requirements, with up to 21,883 ppm DCM in the formulations. **Conclusions:** Overall, the relatively high residual DCM concentration and the suboptimal physical stability do not make the approach easily usable for generating supersaturated lipid-based formulations.

## 1. Introduction

Many new compounds developed today show poor aqueous solubility [1]. This can be especially challenging when administering orally, potentially leading to poor absorption and, therefore, an insufficient pharmacological effect for the patients. For this reason, formulations that can enable the oral administration of those drugs (enabling formulations), such as amorphous solid dispersions, mesoporous carriers, complexing agents, and lipid-based formulations, have been explored and developed [2]. Lipid-based formulations (LBFs) as drug delivery systems are especially interesting for the oral administration of highly lipophilic compounds, so called ‘grease ball’ molecules. LBFs are composed of lipids, surfactants, and/or cosolvents and can be classified depending on the content of each component and the hydrophilicity of the surfactant by the lipid formulation classification system into class I–IV [3,4]. The most lipophilic formulation type, LBFs type I, consists solely of lipids, whereas type IV LBFs consist of surfactants and cosolvents only.

A disadvantage is that the drug load in LBFs is confined by drug solubility in the formulation vehicle, which can be problematic when high drug loads are required, e.g., for preclinical toxicity studies. For this reason, supersaturated LBFs, in which drug concentrations exceeding the equilibrium solubility in the vehicle, have been introduced. In the literature, supersaturation was induced via a heat-based approach. The principle of heat-induced supersaturation has successfully been applied for different supersaturated LBFs and with a range of different compounds [5,6,7,8,9,10,11,12,13]. Apart from liquid supersaturated LBFs, solid supersaturated LBFs have also been described [14,15,16,17,18,19].

The increase in drug solubility with increasing temperature in LBFs is highly compound-specific, as recently demonstrated by Bennett-Lenane and coworkers [20]. From a dataset of 21 compounds, the solubility in type I LBFs (either medium-chain or long-chain based) was reported to increase 1.0–3.4-fold from ambient temperature to 60 °C [20]. Holm and colleagues [21] recently reviewed the current state of supersaturated LBFs and reported that for a temperature increase from ambient temperature to 60 °C, a 1.5–2.5-fold increase in solubility in a LBF can be expected based on the literature. However, there are exceptions to this general estimation [21]. Albeit, for solid supersaturated LBFs, it has been demonstrated that it is possible to fabricate formulations with a far greater drug load, as ibuprofen was reported to be supersaturated in a lipidic excipient at 965% of its equilibrium solubility at ambient temperature; however, crystalline ibuprofen was detected in the formulation [15].

Supersaturation is a thermodynamically unstable state; hence, drug precipitation may occur at some stage. The physical stability of supersaturated LBFs seems to be highly drug- and formulation-dependent. Supersaturated type IIIa LBFs loaded with simvastatin at 150% equilibrium solubility were reported to be stable for 10 months at ambient temperature [22], whereas supersaturated type I and type II LBFs with cinnarizine and celecoxib, with drug loads of up to 138% and 136% of the equilibrium solubility at 37 °C, were reported to precipitate within 14 days in most cases [6].

Apart from inducing supersaturation in LBFs by heat, other approaches have, to the best of our knowledge, not been explored so far. Dissolving a compound and excipients in a volatile organic solvent followed by evaporation of the solvent has been conducted to prepare other types of formulations, e.g., to induce supersaturation in transdermal formulations [23] and amorphous solid dispersions [24], as well as to increase the drug load in lipid nanocarriers [25]. Additionally, solid SNEDDS with mesoporous silica as a carrier [26] were prepared using this technique. Therefore, solvent evaporation for the preparation of liquid supersaturated LBFs could be an interesting approach, potentially resulting in higher supersaturation ratios, and it could also offer opportunities for heat-sensitive compounds. A potential issue may be residual solvents in the drug product, which may be toxic and may affect the permeation, dependent on the solvent [27]. The International Council for Harmonisation of Technical Requirements for Pharmaceuticals for Human Use (ICH) has released a guideline stating acceptable limits of residual solvent concentration in pharmaceutical products, depending on the toxicity of the respective solvent [28]. Dikpati and colleagues [29] recently investigated the amount of residual chloroform and acetonitrile upon the preparation of liposomes, polymer nanoparticles, and lipid nanocapsules and found that it is difficult to bring the levels of residual solvents to a level in line with the ICH-guideline with the manufacturing approaches they evaluated [29].

To explore the possibility of generating higher levels of supersaturation in liquid LBFs, the aim of the current study was to investigate supersaturation induction in classical type I LBFs by the solvent addition approach with dichloromethane (DCM), using celecoxib and fenofibrate as model drugs, which are established in the field of LBFs and possess different physicochemical properties. Furthermore, physical stability and residual DCM were investigated to assess if the approach could be used in a preclinical setting.

## 2. Materials and Methods

### 2.1. Materials

Labrafac^TM^ Lipophile WL1349 (medium-chain triglyceride, MCT) and Maisine^®^ CC (long-chain monoglyceride, LCM) were kindly gifted from Gattefossé (Lyon, France). Dichloromethane (HPLC-grade), tetrahydrofuran, acetonitrile, and trifluoroacetic acid were from VWR (Leuven, Belgium). Celecoxib was purchased from Astatech Inc. (Bristol, PA, USA), and fenofibrate was purchased from Sigma-Aldrich (Bornem, Belgium). 1,3-Dimethyl-2-imidazolidinone was from TCI (Zwijndrecht, Belgium). All chemicals used were of analytical grade. Ultra-purified water was from a Milli-Q^®®^ Advantage A10 Water Purification System (Burlington, VT, USA).

### 2.2. Solubility Studies in Lipid-Based Formulations at Different Temperatures

The BCS-class II drugs celecoxib (weak acid, log P = 4.3, 381.4 g/mol) and fenofibrate (neutral, log P = 5.24, 360.8 g/mol) were selected as model compounds. The solubility of these two compounds was determined in two vehicles resembling type I LBFs, a long-chain monoglyceride (Maisine^®^ CC) and a medium-chain triglyceride (Labrafac^TM^ Lipophile WL1349) at ambient temperature (~21 °C), 37 °C, and at 60 °C, to determine the potential for heat-induced supersaturation. An excess amount of drug compound (celecoxib or fenofibrate, respectively) was added to 1 mL of the lipid vehicle, and the vials were incubated and constantly rotated for 48 to 72 h at ambient temperature, 37 °C, and 60 °C (Weiss Temperature cabinet, Hamburg, Germany). In the case that the added compound was dissolved during incubation, more compound was added. After incubation, the samples were centrifuged in an Eppendorf centrifuge 5430R (Hamburg, Germany) for 30 min at 17,500 rpm (=30,130 rcf) at either 20 °C (samples incubated at ambient temperature), 37 °C (samples incubated at 37 °C), or 40 °C (samples incubated at 60 °C). The supernatant was diluted and further analyzed by UPLC-UV, as described below.

### 2.3. Formulation Preparation Using Dichloromethane

For the preparation of supersaturated LBFs using the solvent-based method, 1000 µL of the respective lipid (either LCM or MCT) and drug compound (celecoxib and fenofibrate, respectively) in an amount equivalent to 100%, 150%, 200%, and/or 250% of the equilibrium solubility determined at 60 °C were dissolved in DCM in a dilution of 1 + 2 and 1 + 4 (1000 µL lipid diluted with 2000 and 4000 µL DCM, respectively) in a 10 mL screw-top vial (dimensions: 22.5 diameter × 46 mm height) and mixed (drug and lipid were completely dissolved). Immediately thereafter, DCM was evaporated in a Genevac EZ2plus Personal evaporator (Sysmex Belgium NV, Hoeilaart, Belgium), under vacuum, using the “HPLC fraction” program at 37 °C for 19–26 h, until the DCM was evaporated as assessed by visual evaluation using a reference with an equivalent amount of the lipid component in a similar vial. The solubility of the drug compounds in the vehicles was determined as described above, and the supernatant from the solubility measurement at 60 °C can be regarded as the respective heat-induced supersaturated LBF with a drug compound load of 100% at 60 °C.

### 2.4. Formulation Characterization

#### 2.4.1. Physical Stability

After preparation in the vacuum evaporator at 37 °C, the supersaturated formulations prepared by solvent evaporation were stored at ambient temperature (~21 °C), and the physical stability was evaluated by visual inspection for precipitation within the timeframes < 1 day, 1–4 days, 4–7 days, 7–14 days, 14–28 days, or > 28 days. For selected samples where precipitation was visible, a few droplets (enough volume to cover the area of a quartz microscope slide covered by a cover slip) were taken from the middle of the vial using a plastic pipette, and the crystallinity was confirmed by using a ZEISS Axio Vert.A1 microscope with polarized light function (Jena, Germany). This was also conducted for the starting material and the celecoxib and fenofibrate powder, in MCTs and LCMs, as shown in the Appendix A.

#### 2.4.2. Determination of the Amount of Residual Dichloromethane

The formulations showing a higher physical stability were chosen to be investigated further by measuring the amount of residual DCM. These formulations were freshly prepared before the measurement. A total of 100 mg of the formulations were dissolved in 2.0 mL 1,3-Dimethyl-2-imidazolidinone in a headspace vial and further analyzed by GC-headspace as described below.

### 2.5. Analysis

#### 2.5.1. Quantitative Analysis of Solubility Samples

The supernatants obtained from the solubility studies were diluted 1:50 with tetrahydrofuran, which was further diluted 1:100 with acetonitrile. This was subsequently diluted into relevant concentrations with an 80/20 (*v*/*v*) acetonitrile/0.1% trifluoroacetic acid-in-water mixture for fenofibrate samples and with a 90/10 (*v*/*v*) acetonitrile/0.1% trifluoroacetic acid-in-water mixture for celecoxib samples. This final dilution was analyzed by using an Acquity (UPLC™) H-class system composed of a quaternity solvent manager, a photodiode array (PDA) detector, a sample manager, and the Empower^®^ software (version 3.0) for data processing. For the chromatographic analysis, a reversed-phase Waters Acquity BEH C18 with a 50 mm × 2.1 mm column (particle size 1.7 μm) including a precolumn (Waters Acquity BEH C18, 5 mm * 2.1 mm, particle size 1.7 μm) (Waters, Milford, CT, USA) was used. The mobile phase consisted of 0.1% trifluoracetic acid in water and acetonitrile in a 40/60 proportion for celecoxib and in a 30/70 proportion for fenofibrate, and isocratic conditions were applied. The injection volume was 4 μL for celecoxib and 2 μL for fenofibrate, and the flow rate of the mobile phase was 0.60 mL/min. The column temperature was kept at 25 °C for celecoxib and at 55 °C for fenofibrate; the detection wavelength was 251 nm for celecoxib and 288 nm for fenofibrate. The method was linear in the range from 0.08 to 15.0 μg/mL for celecoxib (R^2^ > 0.999) and from 0.25 to 25.0 μg/mL for fenofibrate (R^2^ > 0.999).

#### 2.5.2. Residual Solvent Measurement Using GC-Headspace

The sample in the headspace vial was incubated for 10 min at 110 °C in a Combipal Autosampler prior to the injection of a 1 mL sample using a 2.5 mL Headspace syringe (syringe temperature: 150 °C). The Agilent 7890/6890-GC system used for analysis was equipped with an FID detector. A Rxi-624 Sil MS column, with 20 m length and 0.18 mm in diameter, 1.0 µm film thickness, an inlet temperature of 260 °C, and a gas flow (H_2_) of 1.0 mL/min, was used. The samples were detected at 320 °C by FID. The DCM concentration was calculated as ppm (*w*/*w*) according to an external calibration using the average area of six injections of a reference solution, which also contained other residual solvents. Waters Empower software was used for data processing.

#### 2.5.3. Statistical Analysis

The statistical analysis was performed using GraphPad Prism, Version 9.5.1. For the solubility studies, mean and standard deviation (n = 3) was calculated, and unpaired Student’s *t*-tests (*p* = 0.05) were applied to compare the respective solubility values between ambient temperature and 37 °C and between 37 °C and 60 °C for each drug compound in each medium. Additionally, the solubility ratio between 60 °C and 37 °C was compared between the two lipid excipients for each compound using the above-mentioned Student’s *t*-test.

## 3. Results and Discussion

### 3.1. Solubility Studies

Table 1 summarizes the solubility of celecoxib and fenofibrate in two type I LBFs, either based on MCTs or LCMs, at ambient temperature, 37 °C, and 60 °C. These temperatures were selected to reflect normal storage temperature, body temperature, and the temperature normally applied in the literature on supersaturated lipid-based formulations [6,13]. The solubility studies allowed for the determination of the heat-induced supersaturation potential of both drugs in the investigated vehicles. As shown in the Table, the solubility increased with increasing temperature in all cases, with all increases between ambient temperature and 37 °C and between 37 °C and 60 °C being statistically significant. The supersaturation ratio between 60 °C and ambient temperature was 1.85–2.38 for celecoxib and 2.30–2.78 for fenofibrate, with the ratio being significantly higher in LCMs compared to MCTs for both celecoxib and fenofibrate. This range was in line with the literature, with a temperature-induced solubility increase in lipids of 1.0–3.4-fold from ambient temperature to 60 °C, as determined by Bennett-Lenane and colleagues from a dataset of 21 compounds [20]. For celecoxib, the solubility in LCM exceeded the solubility in MCTs at all temperatures. In contrast, for fenofibrate, drug solubility in MCTs was higher than in LCM at all investigated temperatures. The solubility of the two model compounds has previously been determined in medium-chain mixed glycerides (Capmul MCM) and long-chain mixed glycerides (Maisine^®^ CC) at ambient temperature and at 60 °C [6,20], with reported solubilities in the same range as the ones determined in the current work.

### 3.2. Supersaturation Induced with Volatile Organic Solvent and Stability of the Formulations

The degree of supersaturation for the formulations prepared by the solvent method was based on the respective solubility data at 60 °C (Table 1). DCM was chosen as an appropriate solvent to induce supersaturation in type I LBFs, as it was able to dissolve high amounts of the two drug compounds (determined in preliminary experiments) as well as—in contrast to less toxic solvents such as ethanol—the lipid vehicles. Further, the solvent has a low boiling point of 39.6 °C, which enables its removal by evaporation—at least partially. Based on the solubility of celecoxib and fenofibrate in MCTs and LCMs, supersaturated LBFs with drug loads from 100% and up to 250% of the equilibrium solubility at 60 °C were prepared in the two vehicles by adding the compound and DCM followed by evaporation of the solvent. The drug loads relative to the solubility at ambient temperature and the equivalent relative to the solubility at 60 °C are shown in Table 2 for celecoxib and in Table 3 for fenofibrate, and the drug load will be further discussed relative to the solubility at ambient temperature, as at this temperature the physical stability was investigated. The formulations were dissolved in DCM in a 1 + 2 or in a 1 + 4-ratio, i.e., 1000 µL LBF and 2000 or 4000 µL DCM independent on the drug load, in order to investigate the impact of the amount of DCM used for formulation preparation.

After the generation of the supersaturated LBFs and removal of DCM at 37 °C, the physical stability of the preparations was investigated at ambient temperature for both the celecoxib and fenofibrate containing LBFs. Ambient temperature was chosen for the stability studies as this was closer to a real-life scenario, since storage at 37 °C for elongated times was perceived as unusual. The obtained data are shown in Table 2 and Table 3 for celecoxib and fenofibrate, respectively.

As shown in Table 2, celecoxib-loaded supersaturated LBFs with drug loads of 185% and 278% in MCTs and 238% and 356% in LCMs relative to the solubility at ambient temperature did not show any precipitation for 7 days. Formulations with a celecoxib content of 371% or higher showed visible precipitation already within a timeframe of a few days in both investigated vehicles, with one of the formulations already showing drug precipitation during solvent evaporation. Therefore, only the more stable formulations were chosen to be investigated further by quantifying the residual amount of DCM.

For supersaturated MCT-based formulations loaded with fenofibrate, compound loads of 230% and 345% were stable for 7 days, whereas a drug load of 460% relative to the solubility at ambient temperature led to a rapid visible drug precipitation already during solvent evaporation (Table 3). For LCM-based formulations with fenofibrate, all formulations with compound loads of 278% were stable for at least 28 days, whereas when the compound load was 418% or higher, precipitation was visible earlier and occurred erratically (Table 3). Therefore, fenofibrate-loaded formulations with a compound load of 345% or less were investigated further by determining the residual amount of DCM.

Whether the lipid matrix and the drug compound were dissolved in higher or lower amounts of the solvent (1 + 2 or 1 + 4 with DCM, respectively) did not appear to have a clear impact on the physical stability of the formulations after evaporation for neither of the two investigated compounds nor the two investigated vehicles. For some of the formulations prepared via DCM evaporation, the time to precipitation varied between the samples (repetition 1 and 2). This was observed for celecoxib in MCT with a drug load of 371% and for the 1 + 2 and 1 + 4 dilution with DCM; the timeframe in which precipitation was observed was 7–14 days and less than one day, and 1–4 days and 7–14 days, respectively (Table 2). For fenofibrate, this was observed for drug loads of 418% in LCM, with a 1 + 2 and 1 + 4 dilution with DCM. The respective timeframe for precipitation was 14–28 days and less than one day and less than one day and 1–4 days (Table 3). According to the classical nucleation theory, the crystallization process consists of two stages, nucleation and crystal growth. In the nucleation stage, molecules coalesce and reach a critical size of a nucleus. From there on, crystal growth occurs. Nucleation shows poor reproducibility, due to the random nature of nuclei formation, which may explain some of the variation observed in the experiment. Additionally, precipitation may be highly influenced by stimuli, such as scratching the glass wall of the vial. Pictures taken from both celecoxib- and fenofibrate-loaded formulations in MCTs and LCMs under polarized light indicated that the precipitate from the formulations were crystalline (Figure 1A–D). The same applied to the starting material and celecoxib and fenofibrate powder, in MCTs and LCMs (see Appendix A).

These experiments have clearly demonstrated for the first time that a volatile solvent could be used to supersaturate LBFs beyond what has been shown possible by applying heat, for both an intermediate lipophilic and a highly lipophilic compound, namely celecoxib and fenofibrate, respectively. With the evaporation method, it was possible to prepare supersaturated LBFs with up to 2.5-fold higher drug loading when compared to the heat-induced supersaturation method (see Table 1). Similar or even higher drug loads have also been reported for supersaturated silica–lipid hybrids, where Schultz and coworkers reported to have prepared a silica–lipid hybrid loaded with up to 965% ibuprofen compared to the solubility at ambient temperature; although, it has to be considered that crystalline compound was present in the formulation. They report that some of their ibuprofen-loaded solid LBFs were stable for up to 9 months, but formulations with different drug loads were prepared [15]. It must be mentioned that the tendency for supersaturation, stability, and the solubility in lipids at different temperatures seems highly drug-dependent [20].

The most stable formulations prepared via the DCM-evaporation approach did not show any visible precipitation after 28 days. The stability of heat-induced supersaturated LBFs was out of the scope for the current study but is reported in the literature. Ilie et al. investigated the stability of supersaturated type I and type II LBFs, where supersaturation was induced by heat, and reported that many of the celecoxib- and cinnarizine-loaded formulations showed visible precipitation within 14 days [6]. Grüne and Bunjes [30] investigated the stability of heat-induced supersaturated LBFs (prepared at 65 °C and stored at 20–23 °C), either loaded with fenofibrate or other drugs. The formulations consisted of a mixture of MCT and phospholipids, and it was demonstrated that fenofibrate crystallized after nine weeks of storage. Thomas and colleagues [22] investigated the stability of supersaturated simvastatin-loaded type IIIa LBFs, and the formulations were stable for 10 months when stored at ambient temperature, showing no visible precipitation [22]. Overall, in line with what has been reported in the literature, higher supersaturation ratios are expected to result in lower physical stability. The formulations investigated in the present study produced by the solvent evaporation method had a shorter period where they were physically stable. Since the supersaturation ratio in the formulations prepared by solvent evaporation, and consequently the chemical potential in the formulations to drive the crystallization was much higher, the short period of stability was anticipated.

### 3.3. Residual Dichloromethane

The most stable formulations were investigated for their residual content of DCM, namely celecoxib in MCTs (185% and 278% drug load) and LCMs (238% and 356% drug load) and fenofibrate in MCTs (230% and 345% drug load) and LCMs (278% drug load) (Table 2 and Table 3). DCM was chosen due to its low boiling point and high compatibility with both compounds and LBF. Organic solvents must be removed from the lipid vehicles as much as possible before usage in living organisms, which in this study was conducted under vacuum whereby the supersaturation was induced in the investigated LBFs. According to Witschi and Doelker [27], residual solvents in drug formulations can significantly influence various parameters, including the crystallinity of drug molecules, the glass transition temperature, drug dissolution and absorption, and overall stability. These factors may collectively impact the pharmacokinetic behavior of the drug. Specifically, for ethanol, Lennernäs [31] demonstrated that the presence of ethanol–water mixtures in the gastrointestinal tract can enhance drug solubility and therefore lead to dose dumping in certain oral controlled-release hydromorphone products. Additionally, residual solvents may be problematic with respect to toxicity, depending on the solvent and the amount that was administered. As DCM is a class II solvent according to the ICH guideline on residual solvents, it has a permitted daily exposure of 6.0 mg (ICH, 2021). It was therefore deemed necessary to determine the amount of residual DCM in the lipid vehicle after the evaporation process. Besides these considerations, residual DCM in supersaturated LBFs may potentially impact drug solubility and nucleation/growth behavior. According to the ICH guideline Q3C (R8), the maximum acceptable amount of DCM in a pharmaceutical product for human use is 600 ppm [28]. No guidance is provided for preclinical in vivo studies; however, toxicity studies indicate that toxicity is species-specific, with, e.g., an increased risk for malignant tumors in mice, but not in rats and hamsters, at doses of 1000–4000 ppm DCM [32]. Further, when investigating the toxicity of a novel compound, toxicological effects from the vehicle should be avoided to obtain a clear picture of the compound that is being investigated. Therefore, the values mentioned in the ICH guideline should serve as a good indicator for preclinical administration. The residual DCM was determined with GC-headspace, and the results are shown in Table 4.

There was a significant amount of DCM left in the formulations of more than 600 ppm in all cases investigated, spanning from 0.32 to 2.73% of the DCM content added to the LBFs before the evaporation (Table 4). There was a trend that more residual DCM was left in the LCM-based formulations than in the MCT-based formulations. When looking into the effect of extent of dilution (1 + 2 or 1 + 4–dilution), no differences were observed for formulations based on LCM, whereas MCT-based formulations showed a proportional increase in residual DCM with increasing amounts of DCM used during the preparation. MCT-based formulations with higher drug loads of fenofibrate tended to have higher amounts of residual DCM. For celecoxib, this was less pronounced, and there was no impact of the drug load on the residual DCM for LCM-based formulations. In general, formulations loaded with lipophilic fenofibrate (log P = 5.24) tended to have higher residual amounts of DCM than their counterparts loaded with the less lipophilic celecoxib (log P = 4.3). The addition of a more lipophilic compound to the LBFs will most likely generate an overall more lipophilic vehicle, where the highly lipid soluble solvent, here DCM, may have a higher affinity than for the LBF with the less lipophilic compound. In line with these results, Dikpati et al. [29] recently investigated the amount of residual chloroform in lipid films and found that it was very difficult to completely evaporate the solvent. Dikpati and coworkers [29] reported that vacuum drying removed chloroform from investigated lipid films unpredictably; when drying overnight on a vacuum ramp, seven out of the nine investigated samples showed acceptable levels of residual chloroform [29]. Nguyen et al. [25] prepared lipid nanocarriers with a solvent mediated method, utilizing acetone. They removed acetone by rotary evaporation and investigated the residual amount of the solvent over time, reporting a residual acetone content of around 0.1% [*v*/*v*] after 65 min, which could also suggest that it is difficult to remove completely [25].

There are commercially available drug products that are produced based on organic solvents such as liposome preparations dried by spray drying of solutions [33]. Also, several commercially available amorphous solid dispersions are produced by spray drying of organic solvents [34], hence the preparation methods employed ensure control over residual solvent levels, maintaining them below the critical threshold.

Challenges need to be overcome to administer supersaturated LBFs based upon the solvent evaporation method used in this study to different animal species in a preclinical setting, e.g., by using a less polar solvent, such as ethanol, in combination with DCM to lower the solvents affinity. The use of ethanol as a solvent would be interesting for more hydrophilic types of compounds or LBFs than the type I formulations investigated in the present study. Also, the use of a longer evaporation time could be considered, as well as other methods that may remove DCM more efficiently, e.g., conventional solvent evaporation or other evaporation settings. For LBFs, spray drying would not be a suitable approach due to the liquid form of the LBF, but lyophilization could be an option worth considering using suitable solvents.

As the solvent supersaturation method was clearly superior relative to heat induction for LBFs with regard to drug loading, further work is necessary to control the residual solvent content in the supersaturated LBFs. Also, the physical stability of the supersaturated LBFs must be improved before they may become truly relevant to support the logistics around nonclinical studies. Therefore, further research is necessary to control the residual solvent content and to increase the physical stability of the formulations (e.g., by optimization of the drying conditions and by the addition of a precipitation inhibitor). This improved formulation may then be further characterized in a preclinical in vivo model. In vitro, e.g., drug precipitation can be investigated under digestive conditions, as previously shown by Salim et al. [35].

All in all, solvent-induced supersaturation in LBFs still seems to be a promising approach to induce supersaturation in LBFs, given the very high levels of supersaturation that can be reached, which have not been possible using a heat-based approach, but the physical stability and residual solvent content still poses a challenge that needs to be considered in context of the application of the supersaturated LBFs.

## 4. Conclusions

The present study showed that by inducing supersaturation in LBFs loaded with celecoxib and fenofibrate by the evaporation of DCM, compound loads of up to 475% for celecoxib and of up to 557% for fenofibrate compared to the equilibrium solubility at ambient temperature were reached, although crystalline precipitation occurred rather rapidly. The achieved compound loads were up to 2.5-fold higher than the one possible to achieve by inducing supersaturation thermally. With a selected subset of formulations, it was found that the residual amount of DCM after the used evaporation method exceeded acceptable values up to 36-fold. All in all, solvent evaporation was useful in generating LBFs with superior drug loads compared to other methods, but further research is necessary to reduce the residual solvent level and to increase the physical stability, e.g., by considering alternative solvent, preparation methods, and the addition of a precipitation inhibitor.

## Figures and Tables

**Figure 1 pharmaceutics-17-00702-f001:**
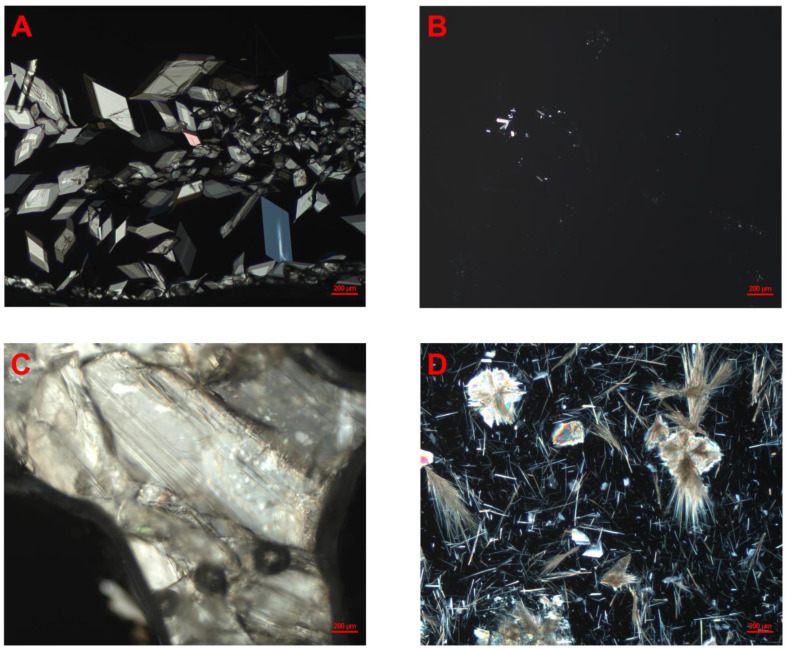
Pictures of celecoxib- and fenofibrate-loaded LBFs under polarized light, observed using optical microscopy. (**A**) Fenofibrate in MCT-based lipid-based formulation, 345% drug load, and 1 + 2-dilution with dichloromethane. (**B**) Celecoxib in MCT-based lipid-based formulation, 278% drug load, and 1 + 4-dilution with dichloromethane. (**C**) Fenofibrate in LCM-based lipid-based formulation, 557% drug load, and 1 + 2-dilution with dichloromethane. (**D**) Celecoxib in LCM-based lipid-based formulation, 475% drug load, and 1 + 4-dilution. All formulations were stored at ambient temperature.

**Table 1 pharmaceutics-17-00702-t001:** Solubility of celecoxib and fenofibrate in medium-chain triglycerides (MCTs) and long-chain mixed glycerides (LCMs) at ambient temperature (~21 °C), 37 °C, and 60 °C, n = 3; mean ± standard deviation. For each compound in both MCTs and LCTs, all increases between ambient temperature and 37 °C and between 37 °C and 60 °C were statistically significant (*p* = 0.05).

	Celecoxib		Fenofibrate	
Temperature [°C]	Solubility MCT [mg/mL]	Solubility LCM [mg/mL]	Solubility MCT [mg/mL]	Solubility LCM [mg/mL]
Ambient temperature	7.72 ± 0.16	10.34 ± 0.08	72.42 ± 0.15	44.04 ± 0.97
37 °C	11.30 ± 0.05	15.57 ± 0.09	144.07 ± 0.55	95.78 ± 2.09
60 °C	14.32 ± 0.25	24.56 ± 0.36	166.51 ± 1.39	122.64 ± 4.13
Ratio of solubility 60 °C/Ambient temperature	1.85 ± 0.03	2.38 ± 0.04	2.30 ± 0.02	2.78 ± 0.11

**Table 2 pharmaceutics-17-00702-t002:** Stability of celecoxib in medium-chain triglycerides (MCTs) and long-chain monoglycerides (LCMs). All investigations were performed as n = 2, except for a formulation based on LCMs with a drug load of 356% relative to the solubility at ambient temperature and a 1 + 4-dilution (n = 1). Formulations marked with ‘*’ were not possible to prepare, as the compound already precipitated during solvent evaporation.

Lipid-Based System	Compound Load Relative to Solubility at 60 °C [%]	Compound Load Relative to Solubility at Ambient Temperature [%]	Dilution with DCM	Repetition	Stability [Days]
MCT	100	185	1 + 2	1	14–28
				2	>28
			1 + 4	1	>28
				2	>28
	150	278	1 + 2	1	14–28
				2	14–28
			1 + 4	1	14–28
				2	7–14
	200	371	1 + 2	1	7–14
				2	<1 *
			1 + 4	1	1–4
				2	7–14
	250	464	1 + 2	1	<1 *
				2	1–4
			1 + 4	1	1–4
				2	1–4
LCM	100	238	1 + 2	1	>28
				2	>28
			1 + 4	1	>28
				2	>28
	150	356	1 + 2	1	7–14
				2	7–14
			1 + 4	1	7–14
	200	475	1 + 2	1	1–4
				2	1–4
			1 + 4	1	1–4
				2	1–4

**Table 3 pharmaceutics-17-00702-t003:** Stability of fenofibrate in medium-chain triglycerides (MCTs) and long-chain monoglycerides (LCMs). All investigations were performed as n = 2, except for two formulations based on LCMs with a drug load of 557% relative to ambient temperature and 1 + 2- and 1 + 4-dilution, respectively. Formulations marked with ‘*’ were not possible to prepare, as the drug already precipitated during solvent evaporation.

Lipid-Based System	Compound Load Relative to Solubility at 60 °C [%]	Compound Load Relative to Solubility at Ambient Temperature [%]	Dilution with DCM	Repetition	Stability [Days]
MCT	100	230	1 + 2	1	14–28
				2	14–28
			1 + 4	1	14–28
				2	14–28
	150	345	1 + 2	1	7–14
				2	7–14
			1 + 4	1	7–14
				2	14–28
	200	460	1 + 2	1	<1 *
				2	<1 *
			1 + 4	1	<1 *
				2	<1 *
	250	575	1 + 2	1	<1 *
				2	<1 *
			1 + 4	1	<1 *
				2	<1 *
LCM	100	278	1 + 2	1	>28
				2	>28
			1 + 4	1	>28
				2	>28
	150	418	1 + 2	1	14–28
				2	<1
			1 + 4	1	<1
				2	1–4
	200	557	1 + 2	1	4–7
				2	1–4
			1 + 4	1	<1

**Table 4 pharmaceutics-17-00702-t004:** Residual dichloromethane (DCM) content of selected celecoxib and fenofibrate formulations in medium-chain triglycerides (MCTs) and long-chain monoglycerides (LCMs). Compound load refers to the equilibrium solubility in the respective vehicle at ambient temperature.

Drug Compound	Lipid Matrix	Compound Load Relative to Solubility at Ambient Temperature [%]	Dilution	Residual DCM after Evaporation (ppm)	Residual DCM (% of DCM-Content Before Evaporation)
Celecoxib	MCT	185	1 + 2	2113	0.32
			1 + 4	3950	0.49
		278	1 + 2	2452	0.37
			1 + 4	4980	0.62
	LCM	238	1 + 2	13,000	1.95
			1 + 4	15,384	1.92
		356	1 + 2	13,771	2.07
			1 + 4	13,808	1.73
Fenofibrate	MCT	230	1 + 2	5494	0.82
			1 + 4	9333	1.17
		345	1 + 2	12,385	1.86
			1 + 4	19,652	2.46
	LCM	278	1 + 2	18,136	2.72
			1 + 4	21,883	2.73

## Data Availability

The raw data supporting the conclusions of this article will be made available by the authors on request.

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
