# Peer review of "Application of Solvent Evaporation to Generate Supersaturated Lipid-Based Formulations: Investigation of Drug Load and Formulation Quality"

_pharmaceutics, 2025, doi:10.3390/pharmaceutics17060702_

Round 1
Reviewer 1 Report
Comments and Suggestions for Authors
This work presents the possibility to obtain lipid-based formulations with high drug loading by inducing supersaturation through addition (and subsequent removal) of an organic solvent. The experimental design is simple but convincing, and the reviewer appreciates the transparency and the critical review of the negative data on residual solvent. However, the following important points must be addressed before advising for publication.
In the abstract: the use of “drug loads” could be confusing, please briefly define this measure.
Data on stability of formulations obtained by heat-induced supersaturation would be a useful benchmark to fully understand the stability of formulations obtained by solvent-induced supersaturation.
Although the approach is interesting, data are not entirely informative. First, only two replicates per formulation (and sometimes one) are too few to provide any statistical power. Such low numerosity, combined with the broad time windows (1-4, 7-14 and 14-28 days), does not provide a real conclusion on the impact of solvent:lipid ratio on stability (as the authors state in lines 261-265). My advice is to add more replicates and use data to describe stability in terms of average days +/- standard deviation before precipitation occurs.
Line 359: form – from?
Author Response
Response to the reviewer comments:
We highly appreciate the comments and input from all three reviewers. We thank the reviewers for their expertise and are confident that this will help to improve the quality of the manuscript. We have responded to the comments in green in the paragraphs below. Changes in the manuscript are also highlighted in green.
Reviewer #1:
This work presents the possibility to obtain lipid-based formulations with high drug loading by inducing supersaturation through addition (and subsequent removal) of an organic solvent. The experimental design is simple but convincing, and the reviewer appreciates the transparency and the critical review of the negative data on residual solvent. However, the following important points must be addressed before advising for publication.
In the abstract: the use of “drug loads” could be confusing, please briefly define this measure.
By "drug load," we refer to the amount of drug that can be dosed with the formulation, i.e. concentration of drug dissolved in the vehicle. In the results section of the abstract, the values are calculated by dividing the amount that can be dosed (in the case of heat-induced supersaturated LBFs, this equals the solubility at 60°C) by the drug solubility in the vehicle at room temperature. This was already mentioned in the abstract, and we have added further clarification in the abstract in the revised manuscript.
Data on stability of formulations obtained by heat-induced supersaturation would be a useful benchmark to fully understand the stability of formulations obtained by solvent-induced supersaturation.
We acknowledge that data on the stability of formulations obtained by heat-induced supersaturation would be valuable. The stability of these specific drugs in these formulations is not known, but comparable literature data is available. It has been reported in the literature that the tendency for supersaturation and stability in supersaturated LBFs is highly drug-dependent (Bennett-Lenane et al., 2021; https://doi.org/10.3390/pharmaceutics13091398). The vehicle itself may also impact stability. For example, Schultz et al. (2018; https://doi.org/10.1016/j.ejpb.2017.12.012) prepared ibuprofen-loaded solid LBFs with high drug load that were stable for up to one month, suggesting that the solid state of the formulation may have a stabilizing effect. Additionally, Ilie et al. (2020; https://doi.org/10.1080/03639045.2020.1721526) prepared cinnarizine- and celecoxib-loaded supersaturated LBF type I and II using the heat-based approach, but with a different lipid vehicle compared to the celecoxib-loaded type I LBF investigated in our work. Ilie and coworkers reported that many of the formulations showed visible precipitation within 14 days. Grüne and Bunjes (2024; https://doi.org/10.3390/ph17030400) investigated the stability of heat-induced supersaturated lipid-based formulations (LBFs) prepared at 65°C and stored at 20 – 23°C, either loaded with fenofibrate or other drugs. The formulations investigated by Grüne and Bunjes (2024) consisted of a mixture of medium-chain triglycerides and phospholipids, and it was demonstrated that fenofibrate crystallized after 9 weeks of storage. Overall, in line with what has been reported in literature, higher drug loading/supersaturation ratios are expected to result in lower physical stability and since the drug loading obtainable with heat-induction is lower, a higher stability compared to the solvent evaporation approach would be anticipated. We have noted the input from the reviewer in the discussion of the revised manuscript.
Although the approach is interesting, data are not entirely informative. First, only two replicates per formulation (and sometimes one) are too few to provide any statistical power. Such low numerosity, combined with the broad time windows (1-4, 7-14 and 14-28 days), does not provide a real conclusion on the impact of solvent:lipid ratio on stability (as the authors state in lines 261-265). My advice is to add more replicates and use data to describe stability in terms of average days +/- standard deviation before precipitation occurs.
We thank the reviewer for the input. Similar approaches to estimate the stability of supersaturated lipid-based formulations, with comparable observation timeframes and even lower numbers of replica are available in the field (see Ilie et al., 2020; https://doi.org/10.1080/03639045.2020.1721526; Grüne and Bunjes, 2024; https://doi.org/10.3390/ph17030400). We ensure transparency by reporting each of the (in almost all cases) two replicates individually. Due to this, n = 2 was considered sufficient. For the purpose to estimate the physical stability, statistical power may not be necessary, in line with other approaches from literature (see Ilie et al., 2020; https://doi.org/10.1080/03639045.2020.1721526; Grüne and Bunjes, 2024; https://doi.org/10.3390/ph17030400). Additionally, we acknowledge that the time to precipitation can vary between samples (lines 315–318), likely due to the random nature of nucleation as described by classical nucleation theory. In addition, stimuli such as scratching the vial wall, can have an impact on crystallization. In the text the reviewer referred to, there is no definitive conclusion implied but rather it is presented as a possibility.
We have ensured that the time perspective is included of the discussion of the revised manuscript.
Line 359: form – from?
This error has been adjusted in the revised manuscript as suggested by the reviewer.
Reviewer 2 Report
Comments and Suggestions for Authors
The manuscript entitled Application of Solvent Evaporation to Generate Supersaturated Lipid-Based Formulations Is Limited by the Amount of Residual Solvent Present in the Formulation." Your work presents a valuable contribution to the field of lipid-based drug formulations. However, before acceptance, the manuscript requires major revisions to address several concerns Below are the important issues that need improvement. Please rewrite the manuscript properly to increase its clarity and coherence before publication.
The title is informative, but it may be overly long and convoluted. Consider simplifying it while preserving the key message.
The introduction gives useful background information but could better highlight the novelty of the solvent-based supersaturation approach compared to previous approaches along with its possible advantages and limitations.”
While the methods are well-detailed, the reason for choosing dichloromethane (DCM) as the solvent over other volatile solvents with lower toxicity might be explored.
The physical stability results are described well, but further attention on the potential effects of temperature changes during storage on precipitation should deepen the discussion.
The discussion on residual DCM levels is thorough, but a more full explanation of feasible techniques to reduced it (e.g., alternate solvents or an extended period drying processes) would be good.
Some tables, such as Tables 2 and 3, are exceedingly detailed, making them difficult to read at a glance. Consider summarizing major findings in a more palatable fashion.
The discussion on ICH Q3C (R8) recommendations is useful, but an explicit comparison of the discovered residual solvent levels with industry norms for lipid-based formulations would strengthen the argument.
While the study primarily focuses on formulation stability, briefly examining the potential influence of residual solvents on in vivo medication absorption and toxicity would add value.
The conclusion effectively summarizes key findings but could be more assertive in recommending future steps, such as alternative solvent choices or further in vivo testing.
Author Response
Response to the reviewer comments:
We highly appreciate the comments and input from all three reviewers. We thank the reviewers for their expertise and are confident that this will help to improve the quality of the manuscript. We have responded to the comments in green in the paragraphs below. Changes in the manuscript are also highlighted in green.
Reviewer #2:
The manuscript entitled Application of Solvent Evaporation to Generate Supersaturated Lipid-Based Formulations Is Limited by the Amount of Residual Solvent Present in the Formulation." Your work presents a valuable contribution to the field of lipid-based drug formulations. However, before acceptance, the manuscript requires major revisions to address several concerns Below are the important issues that need improvement. Please rewrite the manuscript properly to increase its clarity and coherence before publication.
1.The title is informative, but it may be overly long and convoluted. Consider simplifying it while preserving the key message.
The title has been changed as suggested by the reviewer.
2. The introduction gives useful background information but could better highlight the novelty of the solvent-based supersaturation approach compared to previous approaches along with its possible advantages and limitations.”
We appreciate that the reviewer finds the background information we provided useful. In our opinion, some aspects of the possible advantages and limitations are already highlighted in the introduction, but we have made efforts to further emphasize the novelty aspect of our approach in the revised manuscript.
3. While the methods are well-detailed, the reason for choosing dichloromethane (DCM) as the solvent over other volatile solvents with lower toxicity might be explored.
The solvent was selected not only based on its volatility but also on its ability to dissolve the required amounts of lipidic excipient, including Maisine CC (which contains long-chain fatty acids), and high doses of the investigated drug compound. The solubility of the lipid in the solvent was the primary reason why other volatile solvents, such as ethanol, were not chosen. However, this could be an interesting consideration for future work with more hydrophilic types of LBFs. In our approach all was solubilized in the organic solvent (DCM), but supersaturation could also be induced by spiking the lipid-based formulation with a highly concentrated solution in an organic solvent to the lipid phase that could subsequently be solubilized, a method we didn’t investigate in the present work.
We have highlighted this perspective in the revised manuscript.
4. The physical stability results are described well, but further attention on the potential effects of temperature changes during storage on precipitation should deepen the discussion.
The formulations where supersaturation was induced by solvent preparation were prepared at 37 °C, and then the stability was investigated at ambient temperature. This is described in the method section of the manuscript. The storage at ambient temperature would be closer to a real-life scenario, as it is unusual to store formulations at elevated temperature over a prolonged timeframe. However, higher precipitation tendencies due to higher supersaturation ratios are expected at lower temperatures. Additionally, it has to be considered that at high temperatures, chemical instabilities may occur, including those affecting the lipids.
The reviewer’s comment was noted and changes have been made in the revised manuscript .
5. The discussion on residual DCM levels is thorough, but a more full explanation of feasible techniques to reduced it (e.g., alternate solvents or an extended period drying processes) would be good.
We agree with the reviewer and have modified our discussion accordingly (lines 429–438). Feasible approaches to reduce the residual DCM level could be to use a less polar solvent, such as ethanol, in combination with DCM to reduce the solvent's affinity, or eliminating DCM entirely, which would necessitate the preparation of more hydrophilic formulations - an interesting area for future exploration. Additionally, extending the evaporation time or employing more efficient methods, such as conventional solvent evaporation or alternative evaporation settings, could be considered. For LBFs, spray drying is not suitable due to their liquid form, but lyophilization using appropriate solvents could be a viable option.
6. Some tables, such as Tables 2 and 3, are exceedingly detailed, making them difficult to read at a glance. Consider summarizing major findings in a more palatable fashion.
We have implemented changes to make the tables more appealing and easier to read as suggested by the reviewer.
7. The discussion on ICH Q3C (R8) recommendations is useful, but an explicit comparison of the discovered residual solvent levels with industry norms for lipid-based formulations would strengthen the argument.
We agree that it would be beneficial to have an explicit comparison of the discovered residual solvent levels with industry norms for lipid-based formulations. However, to the best of our knowledge, such comparisons do not exist. Supersaturated LBFs have not been prepared via solvent evaporation before, making this a novel approach. For non-supersaturated versions, dissolving the drug compound in the vehicle is typically sufficient for clinical administration. The ICH guideline is nonetheless very useful, because it is applicable to all types of formulations and is used in a regulatory context. Overall, the ICH guideline remains a highly valuable reference due to its applicability across all formulation types and its utilization within regulatory frameworks.
8. While the study primarily focuses on formulation stability, briefly examining the potential influence of residual solvents on in vivo medication absorption and toxicity would add value.
We modified the discussion according to the input from the reviewer.
9. The conclusion effectively summarizes key findings but could be more assertive in recommending future steps, such as alternative solvent choices or further in vivo testing.
The conclusion has been changed as suggested by the reviewer.
Reviewer 3 Report
Comments and Suggestions for Authors
In this study the authors address the effect of residual solvent on the preparation of supersaturated LBFs. Despite this some points should be addressed before publishing.
The title is confused and should be rephrased as follows :Application of Solvent Evaporation to Generate Supersaturated Lipid-Based Formulations: effect of Residual Solvent on the Formulation Quality
Introduction: please add information about advantages and disadvantages of lipids based formulation as drug delivery systems. Moreover, add hint about types of LBFs and supersaturation meaning and importance. What is rationale of selecting fenofiberate and celcoxcib as model drugs
Methods: please add information about the calculation of drug loading percentage. What is the rationale of very high loading percent of celcoxcib and fenofiberate. Moreover, add more details about the stability of LBFs in the term of particles size ,zeta potential and PDI.
Please, at the end of materials and methods add section about statistical analysis indicated software used, methods of data analysis, samples number, data expression SD or SEM and significant limit, p value.
Results and discussion, please discuss the effect super saturation on the stability of LBFs. Moreover , discus the safety of DCM for biological studies.please refine this section and support your work with similar finding and add the contradictory if present. Please, check the figure 1 for resolution and magnification bars. Please, discuss the study limitations and novelty.
Please, check the manuscript misuse of acronyms
Please, check the references list for 2024, 2025 citation dated.
Comments on the Quality of English LanguagePlease check the manuscript for minor grammar errors and syntax.
Author Response
Response to the reviewer comments:
We highly appreciate the comments and input from all three reviewers. We thank the reviewers for their expertise and are confident that this will help to improve the quality of the manuscript. We have responded to the comments in green in the paragraphs below. Changes in the manuscript are also highlighted in green.
Reviewer #3:
In this study the authors address the effect of residual solvent on the preparation of supersaturated LBFs. Despite this some points should be addressed before publishing.
The title is confused and should be rephrased as follows :Application of Solvent Evaporation to Generate Supersaturated Lipid-Based Formulations: effect of Residual Solvent on the Formulation Quality
The input from the reviewer is valuable. We recognize the need to choose another title, as suggested by another reviewer as well. We propose a slightly revised version suggested by the reviewer: "Application of Solvent Evaporation to Generate Supersaturated Lipid-Based Formulations: Investigation of Drug Load and Formulation Quality”.
Introduction: please add information about advantages and disadvantages of lipids based formulation as drug delivery systems. Moreover, add hint about types of LBFs and supersaturation meaning and importance. What is rationale of selecting fenofiberate and celcoxcib as model drugs
Information about advantages and disadvantages of lipid-based formulation as drug delivery systems, as well as hints about types of LBFs and supersaturation meaning and importance, were implemented in the introduction of the revise manuscript, as requested by the reviewer.
Fenofibrate and celecoxib were chosen as these drug compounds are frequently used and are well established model drugs in the field of LBFs, they have a different lipophilicity, with fenofibrate being highly lipophilic and celecoxib having a lower lipophilicity, and fenofibrate is neutral and celecoxib is a weak acid, hence we wanted to vary the physicochemical properties.
Methods: please add information about the calculation of drug loading percentage. What is the rationale of very high loading percent of celcoxcib and fenofiberate. Moreover, add more details about the stability of LBFs in the term of particles size, zeta potential and PDI.
We aimed to explore the potential of the new method compared to the heat-based approach, hence we tried to apply very high drug loads. The calculation refers to the solubility of the drug in the vehicle at ambient temperature, which we have clarified in the revised manuscript as suggested by the reviewer.
Zeta potential and PDI were not determined, as they were not deemed necessary for this study – we worked with the pure lipid systems, which were never dispersed in an aqueous media. The primary goal was to assess whether very high drug loads could be achieved and to estimate the stability of the formulations ‘on the shelf’, as well as their residual solvent content. In cases where precipitate formed, we investigated whether it was crystalline or amorphous, as this can impact oral bioavailability, with amorphous structures being able to redissolve quickly. As shown in Figure 1 from the PLM images, all precipitate was crystalline, and the scaling bar in the Figure allows for a rough size estimation. Notably, the precipitate from fenofibrate resulted in relatively large crystals. However, it appears the reviewer suggested further investigation into dispersed formulations, such as in biomimetic media or during digestion. This would be a valuable next step, involving measurements of PDI, particle size, and Zeta potential. Similar methodologies have been employed by other researchers (e.g., Grüne and Bunjes, 2024; https://doi.org/10.3390/ph17030400), including follow-up studies on precipitation during digestion using in situ low-frequency Raman scattering spectroscopy (Salim et al., 2023; https://doi.org/10.3390/pharmaceutics15071968). Nonetheless, it is imperative to first address the issues observed in the formulations 'on the shelf,' particularly the stability concerns and high residual solvent content, before advancing to subsequent studies.
We have clarified this in the revised manuscript by incorporating the above elements into the discussion.
Please, at the end of materials and methods add section about statistical analysis indicated software used, methods of data analysis, samples number, data expression SD or SEM and significant limit, p value.
We have included information about the statistical analysis in the manuscript, noting that the program GraphPad, Version 9.5.1, was used to determine the mean and standard deviation (n = 3) of the solubility values of celecoxib and fenofibrate in the lipidic excipient. Furthermore, a student t-test was applied to determine statistically significant differences (p = 0.05) between the solubility values at 37°C and 60°C for each drug compound in each medium. The solubility ratio at 60°C and ambient temperature was compared between the two lipidic vehicles for each compound.
Results and discussion, please discuss the effect supersaturation on the stability of LBFs.
The impact of supersaturation on stability is mentioned in the last part of section 3.2., but we agree with the reviewer that it could be more detailed. We have therefore revised this section to provide a more comprehensive discussion.
Moreover , discuss the safety of DCM for biological studies. Please refine this section and support your work with similar finding and add the contradictory if present.
The impact of DCM toxicity on toxicity is mentioned in section 3.3., but we acknowledge that it could be more detailed. We have therefore revised this section as suggested by the reviewer to provide a more comprehensive discussion.
Please, check the figure 1 for resolution and magnification bars.
The resolution of the figure and the scaling bars have been improved in the revised manuscript.
Please, discuss the study limitations and novelty.
This has been implemented in the revised manuscript according to the suggestions from the reviewer.
Please, check the manuscript misuse of acronyms.
We carefully went through the document and found the following acronyms that were used: LBFs, DCM, GC, HPLC, LCM, and MCT. All acronyms were defined when first introduced and have been used in the literature and/or are well established (e.g. Ilie, et al., 2020; https://doi.org/10.1080/03639045.2020.1721526 for MCT and LCM), but we have adjusted the headings to avoid using acronyms there.
Please, check the references list for 2024, 2025 citation dated.
We have included more references from 2023 and 2024 (see reference list). Now, many of the references cited are from 2020 or more recent. We have cited the most relevant literature, and to the best of our knowledge, there have not been significant developments in this area in 2025.
Comments on the Quality of English Language: Please check the manuscript for minor grammar errors and syntax.
We went through the manuscript carefully according to the suggestion from the reviewer.
Round 2
Reviewer 2 Report
Comments and Suggestions for Authors
- In the manuscript text, reference numbers should be placed in square brackets [ ], and in the references section, the references should be in MLA referencing style. Check the author instructions of the journal.
Reviewer 3 Report
Comments and Suggestions for Authors
Accepted